# Multiple-Parallel Morphological Anti-Aliasing Algorithm Implemented in FPGA

**Dihan Ai, Junjie Xue and Mingjiang Wang \***

Shenzhen Key Laboratory of IoT Key Technology, Harbin Institute of Technology, Shenzhen 518000, China
* Correspondence: mjwang@hit.edu.cn

**Abstract:** During 3D rendering by GPU, jagged patterns will appear at the edge of the image due to the lack of sampling points. To increase the continuity of the image, we use an anti-aliasing technique to process these jagged patterns. In this paper, we review the current anti-aliasing techniques and propose a multi-parallel anti-aliasing algorithm based on Morphological Anti-Aliasing (MLAA). Through an experiment and a comparison of the results, we find that our algorithm achieves better anti-aliasing performance at the edge of the image than Multiple Sampling Anti-Aliasing (MSAA) and MLAA by setting more color gradients. Moreover, our algorithm consumes much less time than MLAA in FPGA implementation by performing edge detection of the image, classification of the aliasing patterns, acquisition of the blend weight coefficients, and calculation of the anti-aliased color in three channels of the image simultaneously. In addition, our algorithm also consumes much less memory than MLAA in FPGA implementation by scaling and optimizing the area texture used by MLAA. A comparison of related works indicates that it is more favorable for the proposed algorithm to perform the anti-aliasing of the image than MSAA and MLAA.

**Keywords:** anti-aliasing; multiple-parallel; morphological; FPGA

## 1. Introduction

In the process of GPU 3D rendering, since the computer screen contains a limited number of pixels, we can only use a limited number of pixels to represent the 3D graphics. Due to the lack of sampling points, the edge of the image will contain jagged patterns such as wavy, circular, and flickering patterns, which seriously affect the quality of the image. To improve image quality, many anti-aliasing techniques have been proposed.

The Super Sampling Anti-Aliasing (SSAA) technique is one of them [1–4]; this technique increases the resolution of the image greatly by increasing the number of sampling points of each pixel. However, when this technique is implemented, the fragment shader needs to progress on all sampling points of each pixel, which increases the hardware complexity and power consumption.

To reduce hardware complexity, the Multiple Sampling Anti-Aliasing (MSAA) technique was put forward [5–12]. By counting the sampling points covered by the primitive in each pixel, we can calculate the anti-aliasing color of the pixel according to the coverage of the sampling points and the original color of the pixel. When this technique is implemented, the fragment shader only needs to progress on each pixel, which greatly reduces the hardware complexity compared with SSAA. In addition, since the technique uses the same number of sampling points as SSAA in each pixel, the technique can achieve the same anti-aliasing performance as SSAA.

To further achieve more color gradients without increasing the number of color buffers, the Enhanced Quality Anti-Aliasing (EQAA) technique and Coverage Sampling (CSAA) technique are proposed [13–16]. EQAA and CSAA are two similar technologies implemented for AMD and NVIDIA, respectively. They use more sampling points than the number of color buffers, and by counting the coverage of these sampling points, they can improve the precision of the color with the same size of color buffers.

Unfortunately, however, the aforementioned techniques are only applicable to the real-time 3D rendering pipeline. Although this rendering pipeline can achieve very good anti-aliasing performance, it consumes a large number of hardware resources inside the GPU during rendering, and the GPU will therefore generate high-power consumption. For embedded mobile devices, they have strict power consumption limits, so this high-power consumption rendering pipeline is not suitable for them. To meet the low-power consumption requirements of mobile embedded devices [17], we can use the deferred 3D rendering pipeline [18], and to improve its image quality, some image post-processing techniques are proposed [19].

The Fast Approximate anti-aliasing (FXAA) technique is one of them [20,21]; this technique improves the image quality without complicated calculations through the methods of edge detection and pixel blending. Although this technique occupies very few system resources, the image quality presented by this technique is very poor, and the border of the image still has obvious jagged shapes.

To improve the jagged shape of the image edge, the Morphological Anti-Aliasing (MLAA) technique and the Enhanced Subpixel Morphological Anti-Aliasing (SMAA) technique are proposed. Based on FXAA, MLAA improves the accuracy of pixel blending weight coefficient calculation by further dividing the edge of the image into different aliasing patterns, which also improves the accuracy of the pixel anti-aliasing color calculation results [22–32]. SMAA adds steps such as corner detection based on MLAA to further improve the anti-aliasing performance of the image, but it also further increases the complexity of the hardware [33].

The FidelityFX Super Resolution (FSR) technique is another image post-processing technique proposed by AMD [34]; it contains two steps: one is the Edge Adaptive Spatial Upsampling, and the other is the Robust Contrast Adaptive Sharpening. The first step is to upscale the low-resolution texture to the target resolution by adaptive upsampling, and the second step is to perform adaptive sharpening to further strengthen the edge information. Although this technique can be used for all pipelines, and the consumption of the hardware is very low when implemented, this technique has many flaws in the presentation of the final image—for example, some pixel-level details will be lost.

In addition, the Subpixel Reconstruction Anti-Aliasing (SRAA) technique and the Temporal Anti-Aliasing (TAA) technique are also suitable for deferred 3D rendering pipelines, but unlike FXAA and MLAA, these techniques are placed before the post-processing of the image.

SRAA is a technique that uses subpixel normals and the location information of sampling points to refine the shading results and present images with a better quality [35]. Similar to MSAA, SRAA only requires the fragment shader to progress once on each pixel of a 3D primitive; however, this technique requires more information—such as the depth and normals about the sampling points of each pixel—than MSAA to refine the pixel color. TAA is a technique that allocates the amount of calculation to different frames to realize temporal anti-aliasing [36–39]. It performs a weighted calculation of the color of the current frame and the historical frame through reprojection and rectification of the history frame, which improves the continuity of colors between frames and eliminates ghosting and other phenomena that may appear between frames. However, this technique requires a large number of storage resources to store the colors of the historical frame when it is implemented, and when the historical frame is rectified, it will involve a large amount of time and power.

Deep Learning Super Sampling (DLSS) has also been proposed [40,41]. DLSS is a technique that combines deep learning and anti-aliasing; it contains two steps: one is the training phase and the other is the processing phase, both of which use the same model. In the training phase, the model needs to be trained and optimized with a large number of original images and corresponding supercomputer antialiased images. After the training, the model is sent to the user, and on the user's side, the processing phase begins. In the processing phase, low-resolution images are first generated and then handed over to the

model to process high-resolution images. No matter how the parameters of this model are changed, it only needs to support 60 frames per second to obtain an anti-aliasing performance close to the supercomputer. This technique can achieve a similar anti-aliasing performance to SSAA without the occupation of the image processing resources, but the training and optimization of the model will increase the complexity of the implementation of this technique.

Among the techniques above, although techniques such as MSAA and EQAA can achieve very good anti-aliasing performance, they require a high hardware complexity and power consumption during implementation, so they can only be used in the GPU of the PC. For the GPU in mobile embedded devices, we need to use low-power anti-aliasing techniques suitable for deferred 3D rendering pipelines, such as FXAA and MLAA [20]. However, the anti-aliasing performance improved by these techniques is very poor. Other techniques, such as DLSS, require very complex training and optimization of the image processing model before implementation, which increases the difficulty of technical implementation. To further improve the anti-aliasing performance achieved by the GPU of mobile embedded devices, we propose a multiple-parallel morphological anti-aliasing algorithm and implemented it in FPGA. The algorithm achieves better anti-aliasing performance and reduces the consumption of storage resources by scaling the area texture, which stores the color blending coefficients between pixels. In addition, the algorithm also reduces the time cost of the algorithm by setting more hardware parallelism.

This paper is organized as follows. In Section 2, the principles of the multiple-parallel morphological anti-aliasing algorithm are discussed, including the detection of the image edge, the classification of the aliasing patterns, the acquisition of the blend weight coefficients, and the calculation of the anti-aliasing color. In Section 3, the FPGA implementation methods of the four steps of the algorithm are demonstrated. Section 4 compares the results of this algorithm with MLAA and MSAA. Finally, Section 5 concludes this paper.

## 2. Multiple-Parallel Morphological Anti-Aliasing Algorithm

The multiple-parallel morphological anti-aliasing algorithm (multiple parallel MLAA) mainly includes four steps: edge detection of the image, classification of the aliasing patterns, acquisition of the blend weight coefficients, and calculation of the anti-aliased color. Before demonstrating each step of the algorithm, we first offer a brief introduction to the representation methods of pixels and their edges in Figure 1.

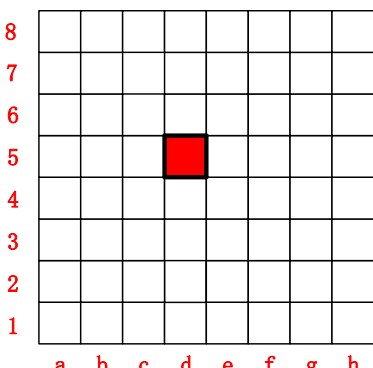

**Figure 1.** Representation methods of pixels and their edges.

We use letters and numbers to represent each pixel and its edges in Figure 1. For example, the red pixels in Figure 1 can be represented as d5, and its four edges can be represented as d5t, d5b, d5l, and d5r, respectively. Each step of the multiple parallel anti-aliasing algorithm is demonstrated as follows.

### 2.1. Edge Detection of the Image

The multiple parallel morphological anti-aliasing algorithm mainly deals with the aliasing of the image edge, so we need to detect the edge of the image first. In our algorithm, the edge detection of an image is performed in a single channel. Therefore, for a color image with R, G, and B channels, we need to perform edge detection in each channel of the image. Next, we will take Figure 2, which only has an effective value in the R channel, as an example to demonstrate the process of the edge detection of the image.

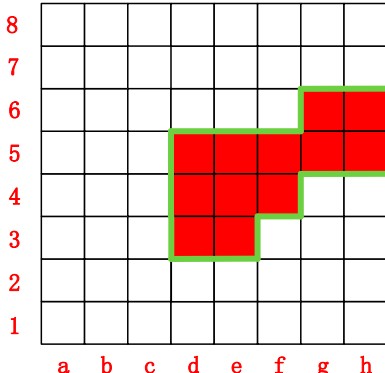

**Figure 2.** Process of the edge detection of the image.

In the process of the detection of the image edge, we traverse each pixel of the image and compare the color value of the currently traversed pixel with the color value of the pixels above and to the left. If the absolute value of the difference between the color value of the current pixel and the color value of the pixels above or to the left is greater than a threshold we preset, then we can determine that the upper edge or left edge of the current pixel is the image edge; otherwise, the upper or left edge of the current pixel is not the image edge. After the traversal, we will obtain the image edge composed of the edges of each pixel. Through this method, we can detect the edge composed of the pixel edge marked by the green line in Figure 2.

### 2.2. Classification of the Aliasing Patterns

After completing the edge detection of the image, we need to classify the edge of the image into different aliasing patterns to facilitate the subsequent calculation of the blend weight coefficient. In our algorithm, since the edge detection of the image needs to be carried out in the R, G, and B channels of the image, we need to classify the aliasing patterns of the edges of the image in the R, G, and B channels. Next, we will take Figure 3, which only has an effective value in the R channel, as an example to demonstrate the process of the classification of the aliasing patterns.

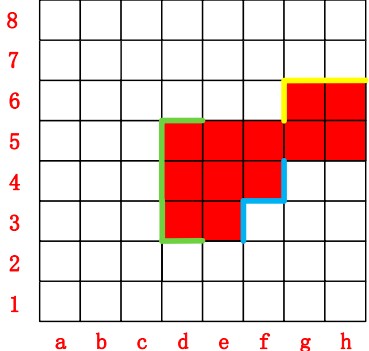

**Figure 3.** Process of the classification of the aliasing pattern.

In the process of the classification of aliasing patterns, we classify the edge of the image into "L-shaped", "Z-shaped", and "U-shaped" aliasing patterns according to the shape of the image edge. Each aliasing pattern is composed of one or two short edges and one long edge, and each aliasing pattern is divided into horizontal or vertical directions according to the orientation of its long edge. After classifying the aliasing patterns of the image edges from the horizontal and vertical directions, we can obtain the horizontal and vertical aliasing patterns of the image. Through this method, after completing the classification of the horizontal aliasing patterns, we can obtain the horizontal "L-shaped" aliasing pattern marked by the yellow line and the "Z-shaped" aliasing pattern marked by the blue line in Figure 3. After completing the classification of vertical aliasing patterns, we can obtain the vertical "U-shaped" aliasing pattern marked by the green line in Figure 3.

### 2.3. Acquisition of the Blend Weight Coefficients

After completing the classification of the horizontal and vertical aliasing patterns of the image, we need to calculate the blend weight coefficients for the pixels included in each aliasing pattern. Similarly, in our algorithm, the calculation of the blend weight coefficients is carried out in a single channel. Therefore, we need to calculate the blend weight coefficient for the pixels included in the aliasing patterns of the image in the R, G, and B channels. Next, we will take Figure 4, which only has an effective value in the R channel, as an example to demonstrate the process of the calculation of the blend weight coefficients.

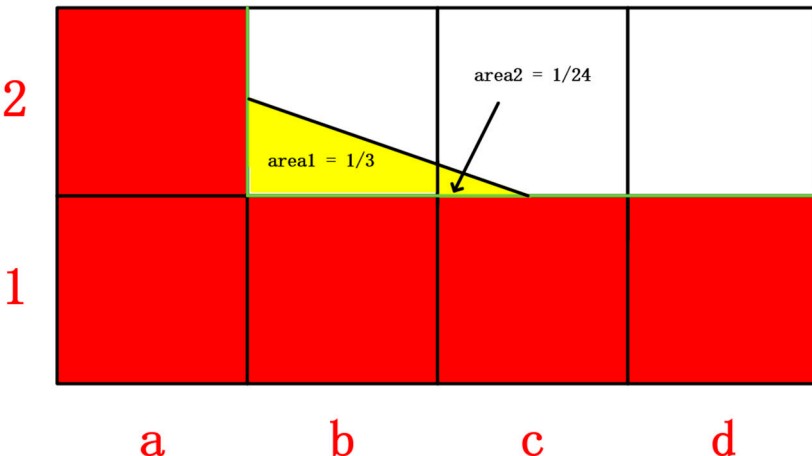

**Figure 4.** Process of the calculation of the blend weight coefficients.

In the process of the calculation of the blend weight coefficients, we first connect the midpoint of each short edge and the long edge in each aliasing pattern to form a crossing edge. Since the crossing edge divides the pixels included in the aliasing pattern, we take the area divided by the crossing edge to each pixel in the aliasing pattern as the weight coefficient for color blending between the pixel and its adjacent pixels. In the image shown in Figure 4, the pixel edge {b2l, b1t, c1t, d2t} marked by the green line constitutes a horizontal "L-shaped" aliasing pattern. By connecting the midpoints of b2l and c1t, we can obtain the corresponding crossing edge marked by the black line. Since the area of the pixel b2 and the pixel c2 is divided by the crossing edge, and they are area1 = 1/3 and area2 = 1/24, respectively, the areas area1 and area2 will be used as the weight coefficients of the pixel b1 and the pixel c1 blended by the pixel above them, respectively, to participate in the calculation of the final anti-aliased color.

To reduce the time and hardware resources consumed when calculating the areas divided by the crossing edge, we predefine an area texture, as shown in Figure 5. The area texture stores the area of each pixel in all aliasing patterns (the 16 aliasing patterns shown in Figure 6) of any length (less than 100) divided by its corresponding crossing edge.

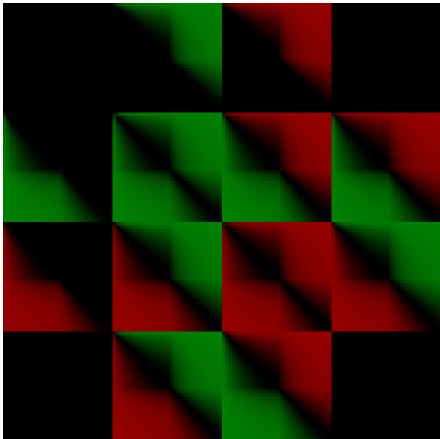

**Figure 5.** Area texture.

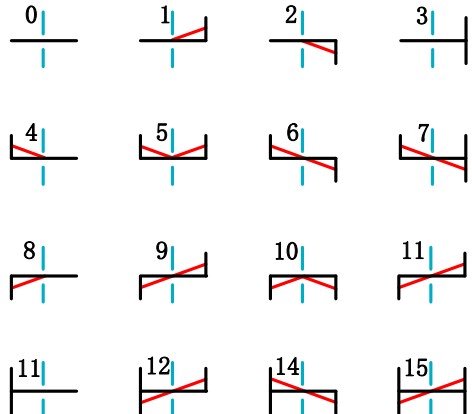

**Figure 6.** Horizontal aliasing patterns.

The definition of the area texture enables us to directly obtain the blend weight coefficients according to the type of the aliasing pattern and the position of the pixel in the aliasing pattern without complicated calculations, which greatly reduces the consumption of time and hardware resources.

### 2.4. Calculation of the Anti-Aliased Color

After obtaining the blend weight coefficient of each pixel in the image, we start to calculate the color of the anti-aliased image. In our algorithm, the color of the anti-aliased image is also calculated in a single channel, so we also need to carry out the calculation of the anti-aliased color in the R, G, and B channels of the image.

In the process of the calculation, we first take the original color of each pixel and then take the weight coefficient of the pixel blended by the color of its adjacent pixels in the horizontal and vertical directions. After calculating the anti-aliased color of the pixel in the horizontal and vertical directions, respectively, we take the average value to obtain the final anti-aliased color of the pixel. Formula (1) is a formula for calculating the anti-aliased color of a pixel.

$$
\begin{aligned}
C_{newX} &= \left(1 - W_{top} - W_{down}\right) \times C_{old} + W_{top} \times C_{top} + W_{down} \times C_{top} \\
C_{newY} &= \left(1 - W_{left} - W_{right}\right) \times C_{old} + W_{left} \times C_{left} + W_{right} \times C_{right} \\
C_{new} &= (C_{newX} + C_{newY}) \div 2
\end{aligned}
\tag{1}
$$

In Formula (1), $C_{old}$ is the original color of each pixel; $C_{top}$, $C_{down}$, $C_{left}$, and $C_{right}$ are the original colors of its adjacent pixels, respectively; $W_{top}$, $W_{down}$, $W_{left}$, and $W_{right}$ are the weight coefficients of each pixel blended by its adjacent pixels; $C_{newX}$ and $C_{newY}$ are

the horizontal and vertical anti-aliased colors of each pixel, respectively; $C_{new}$ is the final anti-aliased color of each pixel.

## 3. Implementation of the Algorithm in FPGA

When implementing the proposed algorithm in FPGA, we found that the edge detection of the image, the classification of the aliasing patterns, the acquisition of the blend weight coefficients, and the calculation of the anti-aliased color can be performed simultaneously in the R, G, and B channels of the image. In addition, the classification of the aliasing patterns and the acquisition of the blend weight coefficients can also be carried out simultaneously in the horizontal and vertical directions of each channel, which improves the parallelism of the algorithm and reduces the time consumption greatly. Figure 7 shows the top-level state machine when the proposed algorithm is implemented in FPGA.

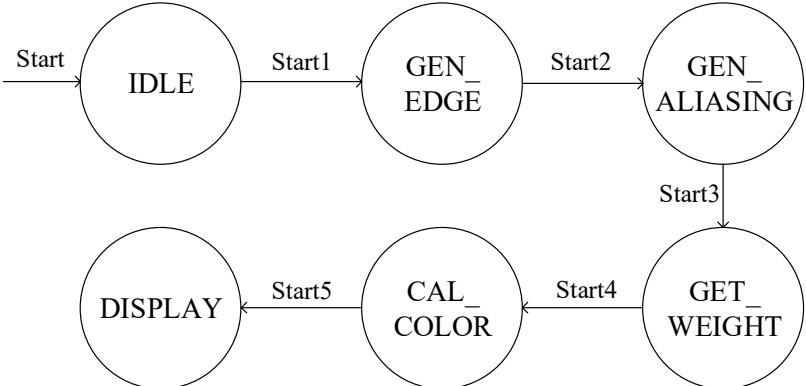

**Figure 7.** State machine on the top level.

It can be seen from Figure 7 that the state machine contains six states, of which state GEN_EDGE, state GEN_ALIASING, state GET_WEIGHT, and state CAL_COLOR correspond to the edge detection of the image, the classification of the aliasing patterns, the acquisition of the blend weight coefficients, and the calculation of the anti-aliased color of the proposed algorithm, respectively.

The state IDLE is the initial state of the system; when the key representing the reset signal rst on the FPGA is pressed, the asynchronous reset signal rst is active (active low), and the system completes the initialization of the internal registers.

When the key representing the start signal Start on the FPGA is pressed, the asynchronous start signal Start is active (active low). At this time, the state of the state machine changes from IDLE to GEN_EDGE, the synchronous signal Start1 is active, and the system starts the edge detection of the image.

After the edge detection of the image is completed, the state of the state machine changes from GEN_EDGE to GEN_ALIASING, the synchronous signal Start2 is active, and the system starts the classification of the aliasing patterns.

By analogy, after the calculation of the anti-aliased color is completed, the state of the state machine changes from CAL_COLOR to DISPLAY. At this time, the anti-aliased image and the original image will be output to an LCD screen connected to the FPGA.

Figure 8 takes the R channel as an example to demonstrate the diagram of the whole system implemented in FPGA. Next, we will describe the FPGA implementation of each step of the proposed algorithm in detail.

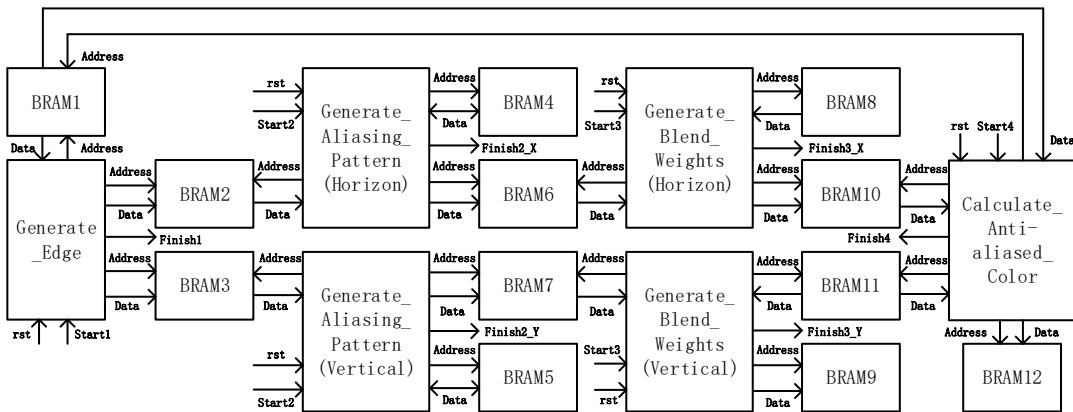

**Figure 8.** Diagram of the whole system implemented in FPGA.

### 3.1. Implementation of the Edge Detection of the Image

The module Generate_Edge in Figure 9 is used to detect the edge of the image. After the module starts to work, we first read the color of the pixel from BRAM1, and then we can determine whether there is a pixel above or on the left side of the current pixel according to its position. If it exists, we read the color of the pixel above or on the left of the current pixel from BRAM1; otherwise, we can determine that the upper edge or the left edge of the current pixel is not an image edge.

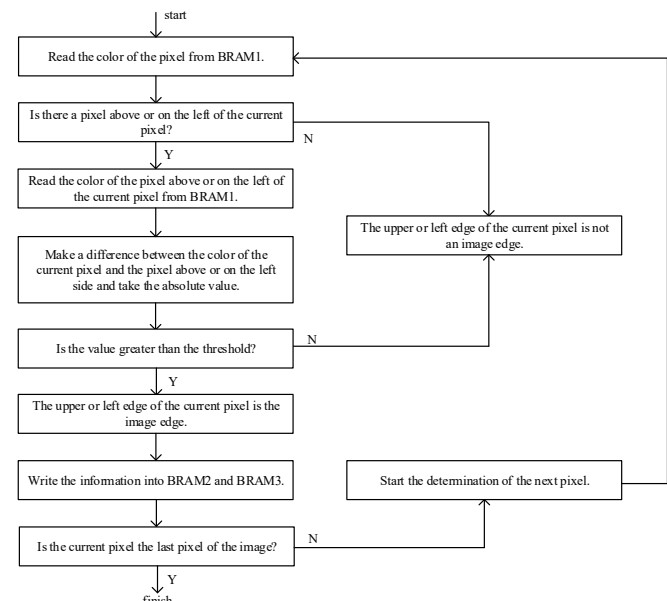

**Figure 9.** Flow diagram of the module Generate_Edge.

After reading the color of the pixel above or on the left side of the current pixel, we note the difference between the color of the current pixel and the pixel above or on the left side and take the absolute value. If the absolute value is greater than our preset threshold, we can determine that the upper edge or the left edge of the pixel is the image edge; otherwise, we can determine that the upper or left edge of the current pixel is not an image edge.

After determining whether the edge of the current pixel is the image edge, we first store the information in BRAM2 and BRAM3, and then we determine whether the current pixel is the last pixel of the image. If the current pixel is the last pixel of the image, it means that we have completed the edge detection of the image; otherwise, we need to continue to determine whether the edge of the next pixel is the image edge.

### 3.2. Implementation of the Classification of the Aliasing Patterns

The module Generate_Aliasing_Pattern in Figure 10 is used to classify the aliasing pattern. It can be divided into horizontal and vertical directions, and we take the horizontal direction as an example to introduce this module.

After the module starts to work, we first determine whether there is a pixel above the left side of the current pixel according to its position. If it exists, we read the mask of the current pixel from BRAM4; otherwise, we can determine that there is no aliasing pattern at the current position. At this time, we can directly start the determination of the next pixel.

After reading the mask of the current pixel, we need to determine whether the current pixel belongs to other aliasing patterns. If it does not belong, we read the edge of the current pixel and the pixel above it from BRAM2; otherwise, we can determine that the current pixel belongs to other aliasing patterns. At this time, we can also directly start the determination of the next pixel.

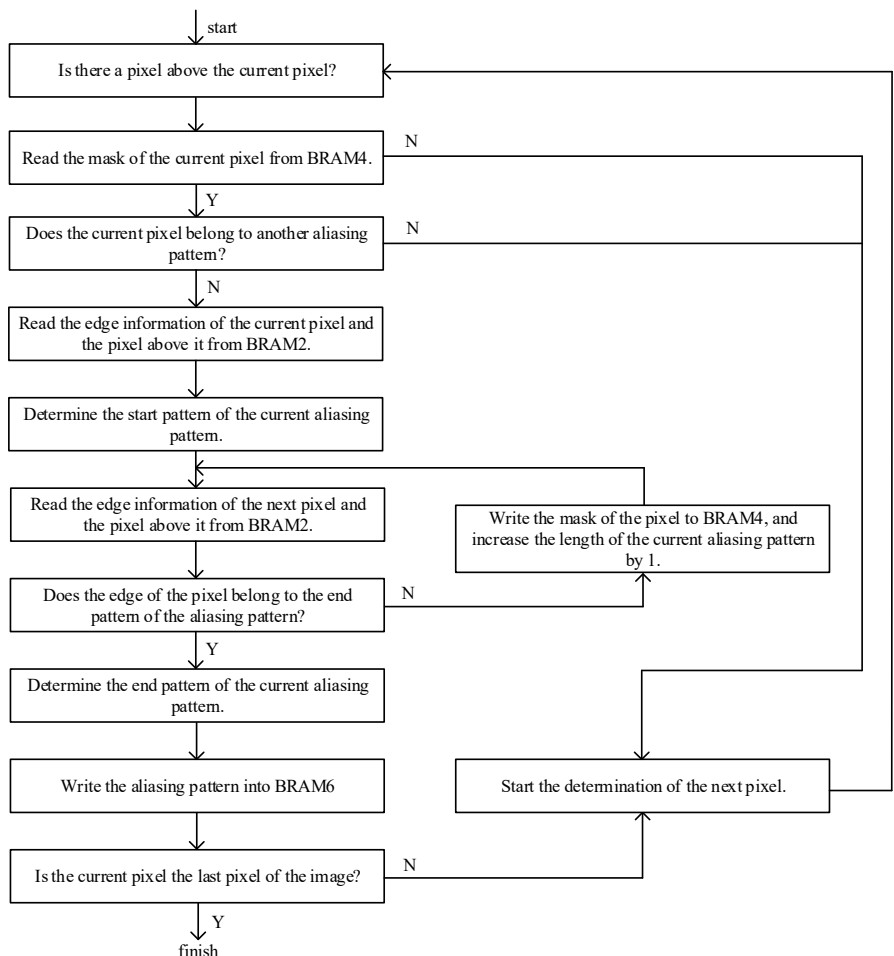

**Figure 10.** Flow diagram of the module Generate_Aliasing_Pattern (Horizon).

After reading the edge of the current pixel and the pixel above it, we can determine the start pattern of the aliasing pattern at this position. After that, we continue to read the edge of the next pixel and the pixel above it from BRAM2 and determine whether the edge is the end pattern of the current aliasing pattern. If it is, we continue to determine the end pattern of the current aliasing pattern; otherwise, we can determine that the edge is not the end pattern of the current aliasing pattern. At this time, we increase the length of the current aliasing pattern by 1, set the mask of the current pixel in BRAM4 to 1, and continue to determine whether the edge of the next pixel is the end pattern of the current aliasing pattern.

After the determination of the end pattern of the current aliasing pattern is completed, we store the start pattern, length, and end pattern of the current aliasing pattern in BRAM6, and then we determine whether the current pixel is the last pixel of the image. If the current pixel is the last pixel of the image, it means that we have completed the classification of the aliasing patterns; otherwise, we need to continue the determination of the next pixel.

### 3.3. Implementation of the Acquisition of the Blend Weight Coefficients

The module Get_Blend_Weights in Figure 11 is used to acquire the weight coefficient of each pixel blended by its adjacent pixels. It can be divided into horizontal and vertical directions, and we take the horizontal direction as an example to introduce this module.

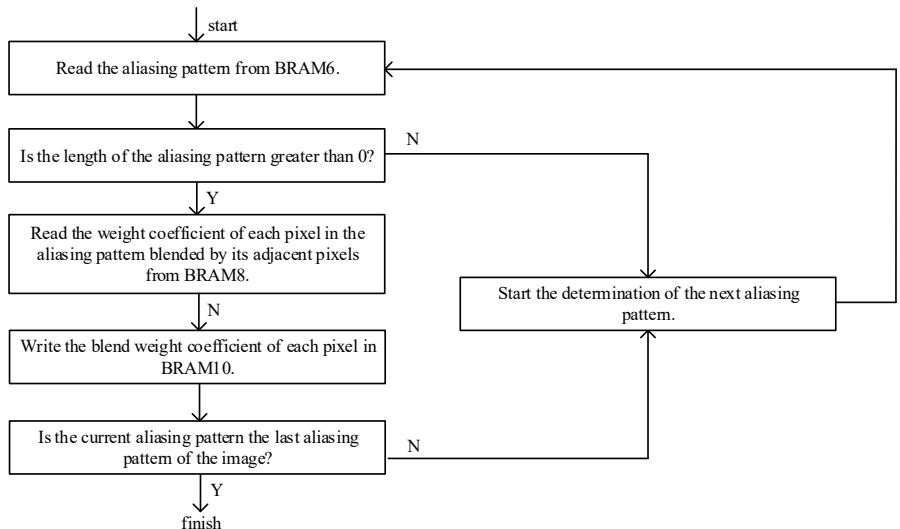

**Figure 11.** Flow diagram of the module Get_Blend_Weights (Horizon).

After the module starts to work, we first read the aliasing pattern from BRAM6, and then we determine whether the length of the aliasing pattern is greater than 0. If it is greater than 0, we need to read the weight coefficient of each pixel in the current aliasing pattern blended by its adjacent pixels from BRAM8; otherwise, we can determine that the current aliasing pattern is invalid. At this time, we can directly start the determination of the next aliasing pattern.

After reading the weight coefficients of each pixel in the aliasing pattern blended by its adjacent pixels, we store these coefficients in BRAM10, and then we determine whether the current aliasing pattern is the last aliasing pattern of the image. If the current aliasing pattern is the last aliasing pattern of the image, it means that we have completed the acquisition of the blend weight coefficients; otherwise, we need to continue the determination of the next aliasing pattern.

### 3.4. Implementation of the Calculation of the Anti-Aliased Color

The module Calculate_Anti-aliased_Color in Figure 12 is used to calculate the anti-aliased color of the image. After the module starts to work, we first read the original color of the pixel from BRAM1, and then we need to determine whether there are pixels around the current pixel. If there are pixels around the current pixel, we read the original color of these pixels from BRAM1; otherwise, we set the variables representing the color and the blend weight coefficients of the pixels around the current pixel to 0.

After reading the original color of the current pixel and its adjacent pixels, we read the weight coefficients of the current pixel blended by its adjacent pixels from BRAM 10 and BRAM 11, and then we can calculate the anti-aliased color of the current pixel according to these coefficients and the original color of the pixels around the current pixel.

After completing the calculation of the anti-aliased color of the current pixel, we store the anti-aliased color of the current pixel in BRAM12, and then we determine whether the

current pixel is the last pixel of the image. If the current pixel is the last pixel of the image, it means that we have completed the calculation of the anti-aliased color; otherwise, we need to continue to calculate the anti-aliased color of the next pixel.

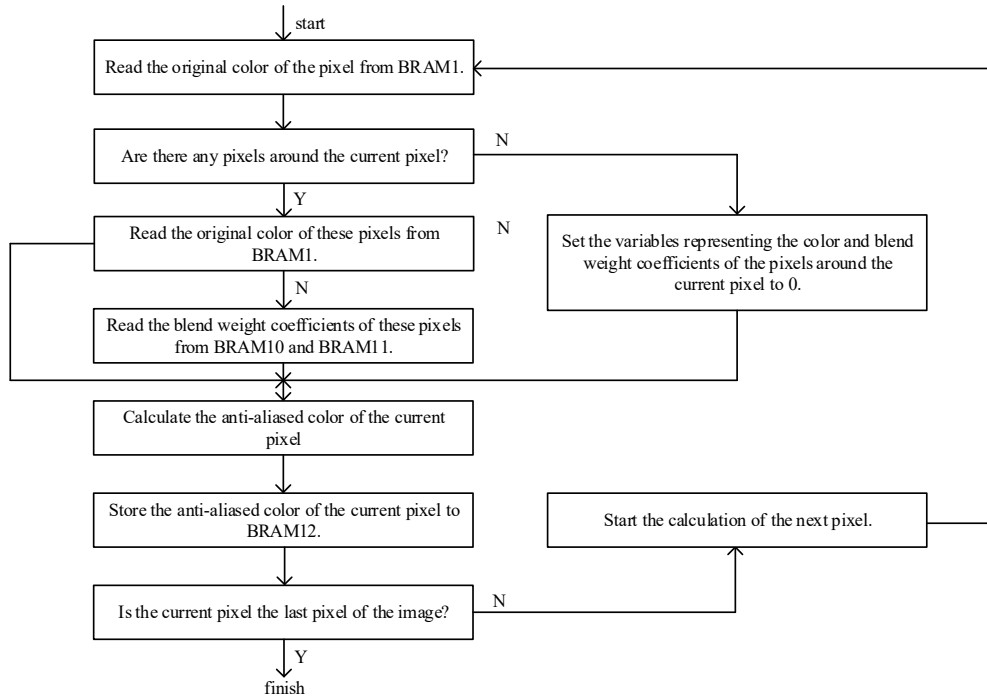

**Figure 12.** Flow diagram of the module Calculate_Aliasing_Color.

## 4. Results and Comparisons

The proposed hardware design was coded in Verilog Hardware Description Language, and it was synthesized in the Xilinx ISE Design Suite and mapped to an FPGA device (xc7z100). The anti-aliasing performances achieved by our algorithm and other algorithms are compared in Section 4.1. After that, the timing performances of our algorithm and MLAA in hardware implementation are compared in Section 4.2. Finally, Section 4.3 compares the memory sizes used by our algorithm and MLAA to store the area texture.

### 4.1. Anti-Aliasing Performance

In this paper, we choose a black-and-white image with a width and height of 100 as the input of our algorithm. It contains a triangle with three vertices: (40, 10), (10, 40), and (90, 25). The anti-aliasing performances achieved by our algorithm and other algorithms are shown in Figure 13.

It can be seen from Figure 13 that there are a large number of obvious jagged patterns at the triangle edge of the original image.

Though MSAA4x can achieve a very good anti-aliasing performance at the boundaries of the image [13,23], it can only achieve very limited color gradients, because MSAA4x corrects the color by calculating the coverage of four sampling points in each pixel, so it can only achieve four color gradients: ×0, ×0.25, ×0.5, and ×0.75. Though we can increase the color gradients of MSAA by increasing the number of sampling points in each pixel (such as increasing the number of sampling points in each pixel to eight to achieve eight color gradients: ×0, ×0.125, ×0.25, ×0.375, ×0.5, ×0.625, ×0.75, and ×0.875), it also increases the hardware complexity when calculating the coverage of each sampling point and the memory size used to store the color of each sampling point.

Although MLAA seems to achieve more color gradients and a better anti-aliasing performance than MSAA4x, the difference between the color gradients is too large, and it achieves a very poor anti-aliasing performance for aliasing patterns of length 1 [22,25].

Our algorithm improves the two disadvantages of MLAA by optimizing the weight coefficient of color blending between pixels. It not only reduces the difference between the color gradients but also improves the processing performance of the aliasing patterns with a length of 1. It achieves a better anti-aliasing performance than MSAA4x and MLAA.

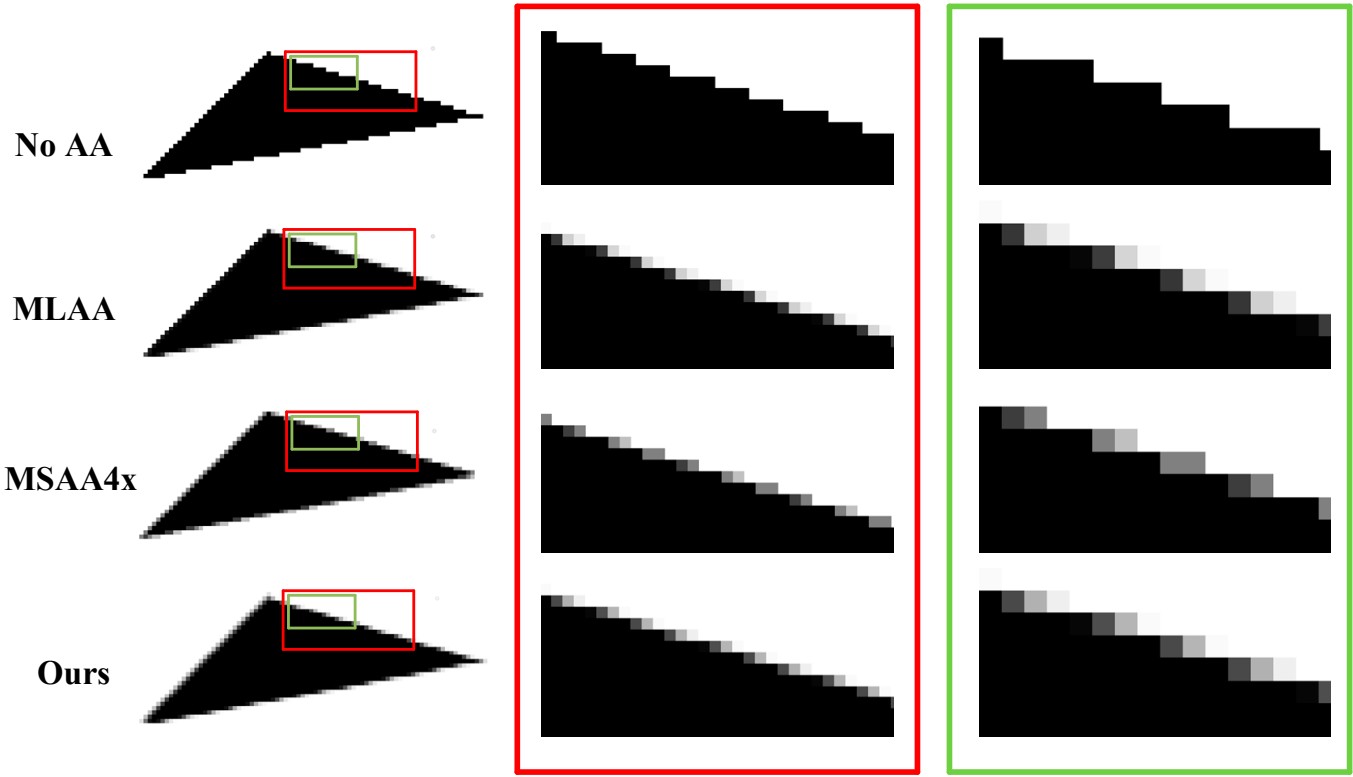

**Figure 13.** Anti-aliasing performance achieved by our algorithm and other algorithms.

*4.2. Timing Performance*

In the process of implementing our algorithm and MLAA in FPGA, we let the system clock of the algorithm and the read & write clock of each Block RAM equal 200 Mhz. We compared the time required for the two algorithms to process images of different sizes and obtain the results shown in Table 1.

**Table 1.** Timing performance of the two algorithms.

| Size of the Image | Ours | MLAA |
|---|---|---|
| Height = 100<br>Width = 100 | 3216.3 us | 11,951.2 us |
| Height = 150<br>Width = 150 | 7214.8 us | 26,817.2 us |
| Height = 200<br>Width = 200 | 12,813.3 us | 47,633.3 us |

It can be seen in Table 1 that our algorithm consumes much less time than MLAA [22,25]. This is because our algorithm performs the detection of the image edge, the classification of the aliasing patterns, the acquisition of the blend weight coefficients, and the calculation of the anti-aliased color for the three channels of the image simultaneously.

In the process of classifying the aliasing patterns and obtaining the blend weight coefficients, our algorithm can also be performed from the horizontal and vertical directions of each channel of the image simultaneously.

We can see in Table 1 that, for the same image, the processing time of MLAA is about 3.7 times that of our algorithm. Thus, compared with MLAA, our algorithm achieves a better timing performance.

### 4.3. Memory Size

In the process of obtaining the blend weight coefficients, we need to store a predefined area texture in the Block RAM of the FPGA. The area texture stores the area of each pixel in all aliasing patterns of any length divided by its corresponding crossing edge. The memory size required to store the area texture is related to the size of the image; the larger the image, the larger the memory required. We compared the memory size required for the two algorithms to process images of different sizes and obtain the results shown in Table 2.

**Table 2.** Memory size used to store the area texture of the two algorithms.

| Size of the Image | Ours | MLAA |
|---|---|---|
| Height = 100 Width = 100 | 100 Kbit | 1600 Kbit |
| Height = 150 Width = 150 | 121 Kbit | 5625 Kbitus |
| Height = 200 Width = 200 | 144 Kbit | 10,000 Kbitus |

It can be seen in Table 2 that our algorithm consumes much less memory than MLAA when they are implemented in FPGA [22,25]. This is because our algorithm scales and optimizes the area texture.

In MLAA, if we choose an image with a width and height of 100 as the input of the algorithm, for each aliasing pattern, we need 10,000 datapoints to store all possible blend weight coefficients. If the width and height of the image were increased to 200, at this point, for each aliasing pattern, we would need up to 40,000 datapoints to store all possible blend weight coefficients. Such a huge amount of data makes it difficult for us to process larger images.

To reduce the amount of data required to store all possible blend weight coefficients for each aliasing pattern, we scaled the area texture used in MLAA. By representing all blend weight coefficients for a certain length interval of each aliasing pattern as a single datapoint, we can greatly reduce the memory size required to store the area texture. For example, we only need 400 and 696 datapoints to store all possible blend weight coefficients for an image with a width and height of 100 and 200, respectively.

We can see in Table 2 that, for the same image, the memory size used to store the area texture of MLAA is about 16 times that of our algorithm when the width and height of the image are 100. When the width and the height of the image increase to 200, the memory size used to store the area texture of MLAA increases to 69 times that of our algorithm. Thus, compared with MLAA, our algorithm reduces the memory size greatly.

### 5. Conclusions

In this paper, we proposed a multiple parallel morphological anti-aliasing algorithm and implemented it in FPGA.

Through experiments and analysis, we find that our algorithm achieves a better anti-aliasing performance at the edge of the image than MSAA4x. This is because our algorithm achieves better continuity at the edge of the image than MSAA4x by achieving more color gradients than MSAA4x. Our algorithm not only reduces the difference between color gradients but also enhances the anti-aliasing processing for the aliasing pattern with a length of 1 by optimizing the blend weight coefficients used by MLAA. Therefore, our algorithm also achieves a better anti-aliasing performance than MLAA.

Moreover, our algorithm consumes much less time than MLAA in FPGA implementation by performing edge detection of the image, classification of the aliasing patterns, acquisition of the blend weight coefficients, and calculation of the anti-aliased color in three channels of the image simultaneously. In the process of classifying the aliasing patterns and obtaining the blend weight coefficients, our algorithm is performed from the horizontal and vertical directions of each channel of the image simultaneously, which also reduces the consumption of time greatly.

In addition, our algorithm also consumes much less memory than MLAA in FPGA implementation. This is because we scale the area texture used by MLAA. By representing all blend weight coefficients for a certain length interval of each aliasing pattern as a single datapoint, we reduce the number of datapoints stored in the area texture, thus reducing the memory required to store the area texture.

Last but not least, although the algorithm proposed in this paper consumes less time than MLAA in FPGA implementation, the latency of the whole system is still too high, because we have not carried out more timing optimization for the hardware implementation of the algorithm, such as pipeline processing. Further timing optimization is required for future research. In addition, the memory consumption of this algorithm is still too large, which makes it impossible for us to further process larger images in FPGA. Thus, memory issues should also be discussed for future research.

**Author Contributions:** Conceptualization, M.W.; methodology, D.A.; software, D.A.; validation, D.A. and J.X.; writing-original draft preparation, D.A.; writing-review and editing, D.A. and J.X.; funding acquistion, M.W. All authors have read and agreed to the published version of the manuscript.

**Funding:** This research received no external funding.

**Conflicts of Interest:** The authors declare no conflict of interest.

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
