# Peer review of "Multiple-Parallel Morphological Anti-Aliasing Algorithm Implemented in FPGA"

_telecom, doi:10.3390/telecom3030029_

Round 1

Reviewer 2 Report

Regardless of the fact that the journal does not have strict formatting requirements, all manuscripts must contain the required sections: A Introduction, Materials & Methods, Results, Conclusions, and the paper does not have all of these, and perhaps that is the reason why it is difficult to understand. The pictures are too big, it is displayed out of the format, so it gives it a visual effect

bad image of work. The number of references in the paper is unacceptably small, so it is no wonder that both State of the art and research gaps are missing.

There is no appropriate validation of the results, no limitations are given for the proposed solutions, and no future work is considered.

Round 2

Reviewer 1 Report

All my comments have been addressed and you have improved your Manuscript. The paper can be accepted.

Reviewer 2 Report

Although the authors tried to remove all the shortcomings given by the suggestions in the previous review, there is still a need for additional minor corrections. Namely, in order for the work to be published in such an eminent journal, there must be a sufficient number of cited references, which, despite the tripled number of references in the work, is still not the case. In particular, references concerning MSA, MLAA and FPGA algorithms are lacking.

Also, citing references would have to be numbered in the order of their appearance in the work, which is a good practice in all important journals, and so that the work could be better followed.
